# Femtosecond-Laser Assisted Surgery of the Eye: Overview and Impact of the Low-Energy Concept

**DOI:** 10.3390/mi12020122

**Published:** 2021-01-24

**Authors:** Catharina Latz, Thomas Asshauer, Christian Rathjen, Alireza Mirshahi

**Affiliations:** 1Dardenne Eye Hospital, D-53173 Bonn, Germany; mirshahi@dardenne.de; 2Ziemer Ophthalmic Systems, CH-2562 Port, Switzerland; Thomas.Asshauer@ziemergroup.com (T.A.); christian.rathjen@ziemergroup.com (C.R.)

**Keywords:** femtosecond laser, fs-assisted cataract surgery, laser-assisted ophthalmic surgery, high pulse frequency, low energy

## Abstract

This article provides an overview of both established and innovative applications of femtosecond (fs)-laser-assisted surgical techniques in ophthalmology. Fs-laser technology is unique because it allows cutting tissue at very high precision inside the eye. Fs lasers are mainly used for surgery of the human cornea and lens. New areas of application in ophthalmology are on the horizon. The latest improvement is the high pulse frequency, low-energy concept; by enlarging the numerical aperture of the focusing optics, the pulse energy threshold for optical breakdown decreases, and cutting with practically no side effects is enabled.

## 1. Introduction

Laser technology and ophthalmic surgery have shaped each other over the past 40 years. The optically transparent structures of the eye, namely cornea, lens, and vitreous body, allow for delivery of the laser energy at different focal depths, thereby giving access to surgical interventions without having to open or mechanically enter the eye (Figure 1). Other types of lasers, with various wavelengths, pulse durations, and power levels, interact with eye tissues in a range of different ways. For continuous laser irradiation of low to moderate average power (mW range), photochemical and thermal effects induced by the absorbed light are the dominant laser–tissue interactions. Depending on the wavelengths used, specific types of molecules can be optically excited to trigger chemical reactions, or local heating of specific tissue can be achieved. If temperatures above 60 °C are reached, tissue coagulation will occur. When pulsed laser light with intensities between 10^7^ and 10^9^ W/cm^2^ interacts with strongly absorbing tissue, near-surface material can be removed explosively. This effect is called “photoablation”. In ophthalmology, it is applied to change the curvature of the cornea with pulsed UV light from excimer lasers. For shorter pulse durations in the ps to fs range and even higher intensities above 10^11^ W/cm^2^, more exotic interactions can be achieved, as will be explained in detail below. A more comprehensive general overview of laser–tissue interaction mechanisms can be found in excellent quality in several text books [1,2].

The first reported ophthalmic use of short pulse lasers at near-infrared wavelengths was in 1979 by Aron-Rosa, who treated posterior capsule opacification (PCO) after cataract surgery [3]. In 1989, Stern et al. demonstrated that by decreasing pulse width of ultrashort-pulsed lasers from nano- to femtoseconds (ns: 10^−9^ s, fs: 10^−15^ s), ablation profiles showed higher precision and less collateral damage [4]. At the same time, optical coherence tomography developed and provided noninvasive three-dimensional (3D) in vivo imaging with fine resolution in both lateral and axial dimensions at a micrometer level [5]. These developments offered ophthalmic surgeons a tool for high precision cutting and visual control through imaging, and ultimately allowed a gamut of treatment applications for these lasers within the field of ophthalmology. Recent changes in the numerical aperture of the laser focusing optics and the repetition rate of the laser sources have further decreased collateral damage while increasing precision. This review article gives an overview of the technical backgrounds of femtosecond lasers and OCT imaging as well as clinical applications in ophthalmic surgery today.

## 2. Laser Technology 

### 2.1. Solid-State Lasers in Ophthalmology

#### 2.1.1. Nd:YAG Laser with Ns Pulse Durations

The first type of short-pulsed laser at near-infrared wavelengths successfully used in ophthalmology was the Q-switched Nd:YAG solid-state laser. Its wavelength of 1064 nm is transmitted by all the visually transparent structures in the eye (cornea, lens, and vitreous body). Their pulse durations are a few nanoseconds (ns), and for ophthalmic applications, pulse energies in the range of 0.3–10 mJ are typically used [6].

When Nd:YAG laser pulses are strongly focused at a location inside the eye, to spot sizes in the order of a few microns, the combination of short pulse duration focusing to minimal spot sizes creates very high intensities at the laser focus, above 10^11^ W/cm^2^. Under these conditions, a phenomenon called “optical breakdown” occurs. In the first step, multiphoton absorption leads to ionization of some tissue molecules, creating free electrons. In the subsequent second step, these “seed” electrons absorb photon energy and are thus accelerated. After repeated photon absorptions, electrons reach a sufficiently high kinetic energy to ionize themselves more molecules by impact ionization, creating more free electrons. If the laser irradiation is intense enough to overcome electron losses, an avalanche effect occurs [2].

When the extremely fast rising electron density exceeds values of approximately 10^20^/cm^3^, a “plasma state of matter” (cloud of ions and free electrons) is created at the laser focus [2]. This plasma is highly absorbing for photons of all wavelengths. Therefore, the rest of the laser pulse is directly absorbed by the plasma, increasing its temperature and energy density (Figure 2).

The hot plasma cloud rapidly recombines to a hot gas, with a thermalization time of the energy initially carried by free electrons of a few picoseconds to tens of ps [7]. This time is much shorter than the acoustic transit time from the center of the focus to the periphery of the plasma volume, leading to confinement of the thermoelastic stresses caused by the temperature rise. Conservation of momentum requires that the stress wave emitted in this geometrical configuration contains both compressive and tensile components [7]. If sufficient pulse energy density is applied, the tensile stress wave becomes strong enough to induce fracture of the tissue, causing the formation of a cavitation bubble [7]. Depending on the pulse energy, the pressure wave can reach supersonic speed a (shock wave). The high plasma temperature also leads to almost immediate evaporation of the tissue within the focal volume, generating water vapor and gases such as H_2_, O_2,_ methane, and ethane [8]. The resulting gas pressure pushes the surrounding tissue further away, adding to the expansion of the short-lived bubble inside the tissue (Figure 2). The maximum volume temporarily achieved by the bubble scales with the pulse energy above the threshold for optical breakdown. During bubble expansion, the inside pressure ultimately drops below atmospheric pressure due to the outward moving material’s inertia, resulting in the bubble dynamically collapsing. The bubble collapse may create another shock wave [2]. This combined process is called “photodisruption” of tissue.

With typical ophthalmic Nd:YAG laser pulse energies, cavitation bubble radii are in the range of 1000–2000 µm, and shock wave amplitudes at 1 mm distance from the focus reach 100–500 bar [10]. These rather pronounced mechanical side effects restrict the use of Nd:YAG lasers. When shorter pulse ps (10^−12^ s) lasers became available, their mechanical side effects proved to be still too large for delicate tasks as required for ophthalmic applications. This limits Nd:Yag laser application in today’s clinical ophthalmological use to cutting isolated tissues, such as the lens capsule in posterior capsular opacification in pseudophakes or small areas of iris tissue to improve aqueous dynamics within the eye.

#### 2.1.2. Femtosecond Lasers

Femtosecond lasers are a more recent advance in solid-state laser technology. They operate at near-infrared wavelengths similar to Nd:YAG lasers but at pulse durations of less than 1 picosecond (ps). As the threshold radiant exposure (J/cm^2^) for inducing optical breakdown in tissue is about two orders of magnitude lower in the fs pulse duration regime than at 10 ns [11], much lower pulse energies can be applied to separate tissue. High pulse repetition rates from 10 s of kHz to even MHz are then used to create continuous cut planes inside the tissue by placing many pulses close to each other with three-dimensional beam scanning systems.

The lower pulse energies lead to a drastic reduction of the mechanical side effects of optical breakdown. For 300 fs pulses of 0.75 µJ energy, the generated cavitation bubbles have radii of only 45 µm, almost two orders of magnitude smaller than ns pulse with energies in the mJ range [12]. In addition, the associated pressure waves are much weaker, 1–5 bar at 1 mm distance [13]. This process is referred to as “plasma-induced ablation”, as the disruptive mechanical side effects of ns pulses described above are absent. Additionally, the thermal side effects of fs pulses in tissue are almost negligible [7].

The first commercially available, USA Food and Drug Administration (FDA)-approved fs-laser system for ophthalmology, the IntraLase^TM^ FS, was launched in 2001 [14]. It was used for “flap” creation in LASIK refractive surgery (see Section 3.1.1 below), replacing mechanical cutting devices called microkeratomes. Its first commercial version operated at a 15 kHz repetition rate and pulse energies of several µJ [15]. Further fs-laser systems for “flap” cutting and other corneal surgery were launched by several manufacturers in the following years, including the Ziemer FEMTO LDV in 2005, which introduced a new concept of low pulse energies and high repetition rates, and later the Wavelight FS200 and the Zeiss VisuMax^TM^.

In 2009, the LensX^TM^ system was introduced, the first commercial fs laser designed for cataract surgery, thus opening a new field of fs-laser application within ophthalmology [16]. Its early versions operated at 33 kHz repetition rate and pulse energies of 6–15 µJ [17]. LensX became part of Alcon, and again, in the following years, multiple other manufacturers launched similar products, including the Johnson & Johnson Optimedica Catalys^TM^, the LENSAR^®^ and the Bausch and Lomb Victus^TM^.

#### 2.1.3. Modern Low Pulse Energy High Repetition Rate Fs Lasers

The pulse energy required to achieve optical breakdown can be reduced in two ways: 

First, by shortening the pulse duration—the latest fs lasers can achieve pulse durations of 200–300 fs, while earlier models had pulse durations of up to 800 fs.

Second, by reducing the focal spot size—the focal volume of a Gaussian laser beam is dependent on the axial extension, the so-called Rayleigh range (z_R_ = πw_0_^2^/λ) and the beam waist w_0_ = fλ/πw_L_, where f is the focal length of the lens, w_0_ the beam radius at the focus, and w_L_ the beam radius at the focusing lens. In other words, the focal volume varies inversely with the cube of the numerical aperture NA = w_L_/f of the focusing optics (Figure 3). The larger the numerical aperture NA, the smaller the focal spot and finally, the smaller the energy threshold for optical breakdown [18].

To practically achieve high NA focusing optics, either the lens diameter can be increased, which quickly becomes bulky and expensive, or the focusing optics can be positioned closer to the eye. The latter approach was implemented by Ziemer Ophthalmic Systems, using a microscope lens with a short focal length as focusing optics and guiding the laser beam via an articulated mirror arm to a handpiece containing the focusing optics, which is docked to the eye at a short distance.

In 2014, the first low pulse energy fs-laser system for cataract and cornea surgery, the Ziemer FEMTO LDV Z8^TM^, was CE-approved and commercially launched. It was more compact and lightweight than its predecessors, enabling mobile use.

### 2.2. Femtosecond Laser–Tissue Interaction

Based on the above laser parameters, the nature of the cutting processes of the two groups differs. In the high pulse energy laser group, the cutting process is driven by mechanical forces applied by the expanding bubbles. The bubbles disrupt the tissue at a larger radius than the plasma created at the laser focus (Figure 4a). On the other hand, in the low pulse energy group, spot separations smaller than the spot sizes are used for overlapping plasmas, which directly evaporate the tissue inside the plasma volume, effectively separating tissue without a need for secondary mechanical tearing effects (Figure 4b). Due to the high pulse repetition rates applied (MHz range), the cutting speeds achieved are similar to the high energy laser group.

The cuts achieved by overlapping plasma evaporation of tissue by low energy pulses, however, have a uniquely smooth surface with virtually no damage to the adjacent tissue [19]. This is important for the quality of corneal “flaps”, lenticules, or also smooth rims of capsulotomy cuts (see Section 3.1.1, Section 3.1.2 and Section 3.3.2 below). High energy pulses with low repetition rate, on the other hand, rely on the mechanical tearing of tissue in between the actual laser foci. This tearing is accompanied by more stress or potentially even damage to the adjacent tissue [20], as shown by the levels of proinflammatory metabolics detected after laser treatments [21,22].

Software arranges the laser spots in the tissue into geometrical patterns. The software also uses scanning systems to position the laser foci in lines, planes, or even 3D geometries. An example of a 3D cut pattern used for cataract lens fragmentation (see Section 3.3.2 below), which combines multiple planes and cylinders, is shown in Figure 5.

The energy of fs lasers with wavelengths in the 1030–1060 nm range is transmitted very well through all transparent structures of the eye. However, opaque material scatters the laser radiation and thus reduces the amount of energy reaching the laser focus. For example, laser cutting the cornea at locations with scars requires higher pulse energies than in normal clear cornea. The energy losses depend on the thickness of the scattering material that the laser light is traveling through before reaching the focus. Therefore, the energy loss is more severe when cutting through a several mm thick nucleus of a cataractous lens than through corneal scars, which are only fractions of a mm thick.

The initial fs-laser systems designed for cataract surgery overcame this by using much higher pulse energies than fs lasers for cornea surgery. In the latest generation of versatile multipurpose ophthalmic fs-laser systems, the pulse energy is adaptable over an extensive range, so that for each situation, the adapted amount of pulse energy can be used, but not more, to minimize side effects, such as excessive gas production.

### 2.3. Supporting Technology Needed in Ophthalmic Fs-Laser Systems

To make an fs-laser device practical for clinical use, some critical supporting technologies needed to be developed as well. Most notable is optical coherence tomography (OCT) imaging of tissue structures, required for the precise positioning of laser cuts deep inside the eye, and the patient interface system using sterile vacuum docking methods to reliably connect the eye to the optical laser delivery system during treatment.

#### 2.3.1. OCT Imaging

OCT is an optical technology that allows for scanning structures inside tissues, thus generating images [23,24]. The images appear similar to ultrasound images but with higher resolution.

The first application of OCT for biological purposes was described by Adolf Fercher et al. for the in vitro measurement of the axial eye length in 1988 (FERCHER 1988). The early clinical OCT systems used so-called time-domain (TD) OCT technology, where the length of the reference arm of an interferometer is mechanically changed. Due to speed limits of this process, these early devices were limited to one-dimensional scans (A-scans), or later small time consuming 2D scans. The so-called frequency-domain OCT (FD-OCT) technology meant a technological breakthrough—it used a fixed reference arm length but a spectrometer with a linear detector array instead of a single detector. Optical path length differences between the interferometer arms in this case produce a periodic modulation in the interference spectrum. By Fourier transformation, an entire A-scan can be retrieved from the measured spectrum [2]. FD-OCT enabled much higher scan speeds, making 2D-imaging and even 3D-imaging feasible in clinical ophthalmology. The first ophthalmic application of FD-OCT, also known as “Fourier domain”, was published in 2002 [25].

Later, a further improved variation of frequency-domain OCT technology was developed, “swept-source” (SS) OCT. In this case, a tunable light source with a frequency sweep indicated by a “sawtooth” frequency profile over time is used in combination with a fast single-pixel detector instead of a spectrometer. For further details of OCT technology, and advantages and limitations of its different versions, Section 7.3 of the textbook by Kaschke et al. [2] provides a comprehensive overview and additional literature references.

The initial ophthalmic use of OCT was exclusively for retinal imaging. Starting in 1994, the technology was also developed for imaging the anterior segment of the eye [26]. The possibility of quickly creating high-resolution cross-section images of the cornea, anterior chamber, and lens was a prerequisite for practical cataract surgery laser systems. Imaging and OCT guided surgery was first envisioned by Zeiss and first demonstrated for femtosecond laser surgery by H. Lubatschowski et al. [27].

In most modern cataract fs-laser systems, three-dimensional OCT scans are performed after docking the laser interface to the eye. The LensAR system uses a different technology, a proprietary 3D confocal structured illumination combined with Scheimpflug imaging [28]. In both cases, the resulting images are then analyzed by image processing software, identifying the tissue boundaries of interest [29]. These are notably the anterior and posterior sides of the cornea, the anterior and posterior surfaces of the lens, and the iris (see Figure 6).

This information is used to automatically propose the suitable positions inside the eye for the planned laser cuts, which are also displayed on screens for checking and confirmation by the eye surgeon (Figure 6).

#### 2.3.2. Vacuum Docking Interfaces

For some laser systems, the patient’s head is placed under a gantry containing focusing optics at a sufficiently long distance to allow the patient’s head to move in and out. In other systems, an articulated arm with a handpiece with focusing optics at its end is used. Due to the flexible arm, the optics can be moved very close to the eye (Figure 7).

The eye’s actual contact is established via sterile, single-use parts, so-called “patient interfaces”. Two different types are in use: (a) applanating interface with a curved or flat interface directly touching the cornea, and (b) liquid-filled interface, where a vacuum ring creates contact to the sclera or the outer cornea, and the center is filled with liquid. The liquid-filled interface allows laser energy transmission while leaving the cornea in its natural shape (Figure 8) [30]. Although contact interfaces temporarily change the shape of the cornea [31], the mechanical contact stabilizes the cornea during surgery to a high degree. This is of particular importance in refractive surgery where precise cuts are required and tissue displacement on a micrometer level has to be avoided. With the absence of clear clinical drawbacks in refractive surgery [32,33,34], contact interfaces will play a dominant role in the future in corneal surgery. Liquid-filled interfaces with little disturbance of the eye might turn out to be the preferred solution in cataract surgery.

The stability of the vacuum docking contact during laser emission is of primordial importance. Loss of contact harbors the risk of cutting in wrong planes. Therefore, all lasers are designed to automatically monitor vacuum levels, sometimes complemented with imaging of the eye position (eye tracking), and to immediately stop laser emission upon loss of contact. Of course, the eye surgeons also monitor their patients during the procedure and can manually interrupt or temporarily pause the treatment when they anticipate problems. In case of laser systems with an articulated arm, the surgeons can also use their substantial manual skills to actively stabilize the laser handpiece while in contact with the eye. In any case, after a vacuum loss, the treatment can usually be resumed immediately after a new docking.

## 3. Clinical Applications

### 3.1. Refractive Surgery

The human eye functions like the lens of a camera. Images are focused on the retina through a converging system composed mainly of the cornea. If the corneal curvature and thus its refractive power does not precisely match the axial length of the eye, refractive problems like near-sightedness (myopia) or far-sightedness (hyperopia) ensue (Figure 9). 

Refractive surgery consists of either reducing the refractive power of the cornea (by flattening) or by increasing its power (by steepening) or by modifying its curvature on a determined meridian to correct astigmatism (cylindrical correction).

#### 3.1.1. Fs Flap Creation for Refractive Surgery

LASIK

In the laser in situ keratomileusis (LASIK) procedure, a corneal flap is created. The flap is lifted and then excimer- or solid-state UV-laser energy is used to change the cornea’s refractive power by flattening or steepening the stromal bed. Later, the flap is repositioned. Before the advent of fs-laser technology, the flap was created using mechanical devices called microkeratomes. With fs-laser technology, the flap can be completed in various patterns (Figure 10). Kezirian et al. compared fs-(IntraLase) created flaps to flaps with two different microkeratomes. They found in the fs group more predictable flap thickness, better astigmatic neutrality, and decreased epithelial injury [35]. Chen et al. confirmed the superiority of fs-laser-created flaps over those cut by microkeratomes. Therefore, in recent years, fs technology has superseded microkeratomes in preparing flaps for LASIK [36]. 

Stromal keratophakia (additive refractive surgery)

Keratophakia as a means to sculpt corneal curvature by adding tissue has been studied since 1949 by Barraquer [37]. Because the quality of the cuts was inconsistent and reactive wound healing along the edges of the cut created additional scarring, it was largely abandoned. With advancements in femtosecond technology, new steps are being taken in the direction of keratophakia. For one, it is now possible to prepare an intrastromal pocket or stromal bed with greater precision. Secondly, new inlay materials are being developed. Current research is focusing on decellularizing and preserving extracted lenticules from lenticule extraction surgeries [38]. 

#### 3.1.2. Intrastromal Pockets

Multiple refractive surgery methods use fs-laser cuts to create “pocket”-shaped openings in the cornea, from which either material can be removed or implanted. In both cases, the refractive power of the cornea changes.

Corneal stromal lenticule extraction

While many fs-associated surgical interventions in ophthalmology are merely improvements of pre-existing techniques, lenticule extraction is unique to fs-laser technology: the procedure was introduced in 2011 to treat myopia, and later also myopic astigmatism. It became known under the brand name “SMILE” (small incision lenticule extraction) of the Carl Zeiss Meditec AG. Later, other companies introduced their own laser systems for similar lenticule procedures under different brand names, including “SmartSight” by Schwind and “CLEAR” (corneal lenticule extraction for advanced refractive correction) by Ziemer Ophthalmic Systems AG.

The procedure is a “flapless” laser refractive technique that uses a single femtosecond laser system to create a pocket. The content of the pocket—the lenticule—is removed via a small access tunnel incision, and as a result, the cornea is flattened (see Figure 11). Instead of an almost 360-degree side cut, as in Lasik, lenticule extraction requires only a small arcuate cut of 50 degrees. Thereby more of the corneal nerves and Bowman layer remain untouched. In addition, sculpting the lenticule instead of ablating the same amount of tissue requires less laser energy. Therefore, the potential advantages of the lenticule technique over traditional laser in situ keratomileusis (LASIK) include reduced iatrogenic dry eye, a biomechanically stronger postoperative cornea with a smaller incision, and reduced laser energy required for refractive corrections [33,39,40,41,42,43]. However, the lenticule procedures have a steeper learning curve for surgeons, with potential complications related to lenticule dissection and removal, limitations with enhancements, and slower visual recovery in the initial phase (three months) [41]. Today, laser-refractive correction of hyperopia is not yet possible with lenticule extraction, but research in this field is ongoing. In a prospective, randomized paired-eye study, SMILE demonstrated good refractive outcomes in terms of predictability, efficacy, and safety. Since LASIK is reportedly an extremely safe and predictable procedure, it is unlikely to prove superiority with alternative methods, such as SMILE [44].

Intrastromal corneal ring segments

Fs technology allows creating stromal pockets of specific size and shape at specific positions. Corneal ring segments (Figure 12) are placed into these pockets to change the curvature of the cornea, specifically in patients with thin and malleable corneas such as in keratoconus, a disease in which the central cornea becomes progressively deformed. Combining this procedure with a tissue strengthening intervention such as corneal crosslinking has been shown to improve uncorrected visual acuity in those keratoconus patients who do not tolerate contact lens correction [45]. 

#### 3.1.3. Intrastromal and Trans-Stromal Cuts for Astigmatic Correction

The concept of corneal cuts for astigmatic correction was established more than 100 years ago [46] and underwent standardization in the late 1980s and 1990s [47,48]. Despite nomograms, there remained a significant level of unpredictability of the manually performed surgery. Astigmatic correction through toric intraocular lenses or corneal ablative surgery largely replaced correction through corneal stromal cuts. Fs-technology offers new opportunities to correct corneal astigmatism by means of corneal cuts: position, length, depth, curvature and the keratotomy angle can be put into practice with unprecedented precision and control. In addition, the fs-specific option of purely intrastromal keratotomies, decreases the potential risk of infection through gaping wounds. Although the general belief is that intrastromal cuts have less effect than transepithelial cuts, there are not enough data published for this relatively young technology to give evidence on significant differences in effectiveness between intrastromal and transepithelial cuts [49]. In general, fs-laser astigmatic correction is possible for both: smaller degrees of astigmatism in healthy corneas and larger astigmatic error in eyes with corneal pathology [50,51].

### 3.2. Corneal Surgery

#### 3.2.1. Penetrating Keratoplasty

Background

Keratoplasty (cornea transplantation) ranks among the oldest and most commonly performed human tissue transplantation types worldwide [52]. A corneal button from a deceased donor is sutured into the recipient cornea. Astigmatism is the leading cause of poor visual outcome after keratoplasty. The better the trephination (cut to separate the corneal button from the cornea) of donor and recipient, the better the fit between the transplant and the recipient and the least the astigmatism. 

Trephination

A perfect trephination system produces a congruent recipient bed and donor buttons and thereby allows well-centered tension-free fitting, and efficiently waterproof-adapting incision edges [53]. Different trephination systems are currently available: handheld, motor-trephine, excimer-laser, or fs-laser based. Comparison of motor-trephine and excimer-based trephination has shown better alignment of the graft in the recipient bed after excimer laser trephination [54]. 

It is often problematic to ensure proper centration with trephination in the recipient eye. Fs technology allows for perfect limbal oriented centration through OCT-visualization.

Another problem with trephination is the mechanism by which the recipient eye and donor button are fixated and stabilized; any mechanical impact on the tissue during trephination causes compression and distortion and will decrease the fit of recipient and donor (Figure 13). Common fixation mechanisms include vacuum and applanation and a combination of both (vacuum suction with applanation). While fs technology avoids some of the trephination pitfalls of mechanical trephination, comparison of fs- and excimer-assisted trephination showed nevertheless superiority of alignment in all sutures-out keratoplasty patients in the excimer group [55].

Different stabilization systems could explain this superiority. While the excimer-assisted keratoplasty does not require applanation of the cornea, it is needed for the fs laser used in the cited studies. The new liquid optics interface assisted fs-Keratoplasty method could solve this problem: Here, cutting both recipient and donor is achieved within a liquid interface, which leaves the curvature of the cornea undisturbed. This reduces shear- and compression artifacts in the tissue and improves congruent fitting of the recipient and donor [56]. It will therefore be interesting to compare liquid optics interface fs-trephinations with excimer laser-assisted trephinations in the future. 

Sidecuts

In femtolaser-assisted keratoplasty (FLAK), different side-cut profiles can be chosen (Figure 14). Theoretical advantages include increased wound surface and thereby accelerated healing and wound stability, better vertical and horizontal alignment of the recipient and donor [57], preservation of healthy recipient corneal endothelium (mushroom), or transplantation of proportionally more endothelial cells with the top hat profile [58,59]. It remains to be seen if other factors, such as suture techniques, have to be modified to transmit these theoretical advantages into true clinical benefits [59]. 

#### 3.2.2. Lamellar Keratoplasty

Background

The cornea is structured in five parallel layers. Often, not all layers of the cornea are diseased. Scars from trauma or infection commonly involve the anterior layers (Bowman layer, anterior stroma). In contrast, some inherited corneal dystrophies (i.e., inborn progressing tissue degeneration) affect only the inner most layers (Descemet’s membrane and endothelium, see Figure 15). Selectively transplanting the pathological layers has several advantages: less tissue is transplanted, and thereby rejection is limited. With the scarcity of donor material, a donor button can theoretically be divided between two recipients. The integrity of the eye is less constrained. Since there is little adhesion between the interfaces of the corneal layers, manipulation at these levels is possible and visual results are excellent. 

Deep anterior lamellar keratoplasty (DALK)

In deep anterior lamellar keratoplasty (DALK), approximately 95% of the anterior corneal layers are removed, and only the innermost layers, Descemet’s membrane, and endothelial cell layer stay behind [60]. It is possible to separate Descemet’s membrane (DM) from the anterior stromal layers by air injection [61]. The surgical difficulty consists of finding the right entry-level for the air injection to initiate separation: too deep and DM is perforated, and the surgery has to be converted to a penetrating full thickness keratoplasty; too high and the air injection will not separate the layers, because only at the true interface is there minimal adhesion and the layers can be separated. Trials to create an fs-assisted cut in the pre-Descemet’s stroma instead of separating the two layers led to lower visual clarity in comparison to true “layer separation” [62]. A new approach in fs technology resolves this dilemma—using OCT visualization, an fs-prepared channel is created that guides the cannula to the desired position and depth of the cornea (Figure 16). The surgeon can control the depth of the injection site individually adjusted to the thinnest point of the patient’s cornea [63]. Buzzonetti et al. compared fs-DALK to mechanical DALK in 20 pediatric patients [64] and concluded that fs-assisted trephination could reduce the postoperative spherical equivalent amount. In conclusion, fs-assisted DALK can improve the success rate of big-bubble creation. By improving donor/recipient fit through fs-created sidecuts, the postoperative spherical equivalent is reduced, and healing accelerated.

Posterior lamellar keratoplasty

Posterior lamellar keratoplasty has revolutionized corneal transplant surgery in the past two decades: the cornea’s clarity depends on healthy endothelial cells. These cells are thought of as non-regenerating highly specialized cells. In a disease called “Fuchs endothelial dystrophy”, but also after traumatic or multiple surgical interventions, these cells cease to do their job in maintaining corneal clarity and patients eventually become blind. Transplanting these cells, be it with a small amount of corneal stroma, so-called Descemet stripping endothelial keratoplasty “DSEK”, or with Descemet’s membrane alone as the carrier, so-called Descemet’s membrane endothelial keratoplasty “DMEK”, reverses the process of corneal opacification and especially in the case of DMEK causes perfect visual acuity [60,65]. Part of the surgery consists of removing the old, nonfunctioning Descemet’s membrane from the recipient cornea. This process is usually done by scraping and pulling the membrane manually. Sorkin et al. have published several papers on creating the descemetorhexis (cutting out of a part of the Descemet’s membrane) with femtosecond laser assistance. The advantage is the perfect centration, shape, and size of the removed area [66,67,68]. It remains unclear if this advantage can solely be attributed to the fs-assisted descemetorhexis or if other causes, such as a better fit of transplanted and remaining Descemet’s membrane, can explain these results. Nevertheless, it shows the immense versatility and breadth of applications that fs-laser technology provides in corneal surgery.

### 3.3. Cataract Surgery

Ophthalmic surgeons have been trying to implement laser in cataract surgery for decades. Bille and Schanzlin proposed ultrashort laser pulses for ablation of the cataractous lens in 1992 [69]. Nagy was the first reporting on the use of fs laser for cataract surgery [16]. There has been a quick evolution of the technology and platform ability since then by several manufacturers. Femtosecond laser technology reduces energy and precise laser application at a certain depth in tissue with minimal damage to adjacent areas. Currently, modern and commercially available fs systems allow the following steps to be taken over by the machine: (a) imaging and measurement of the anterior segment of the eye (incl. cornea, anterior chamber, iris, lens), (b) planning of fs-laser cut application to the tissue (including location depth, pattern, and size), (c) corneal incisions (full thickness for the introduction of instruments to the eye or partial thickness for treatment of corneal astigmatism), (d) circular incision to the anterior lens capsule (capsulotomy), and (e) fragmentation of the cataractous lens nucleus. For all of the abovementioned purposes, the eye must be fixed to laser optics by vacuum docking for precise laser application to the intended area and depth. While some systems use liquid optics interfaces (Ziemer Femto LDV), others have a curved applanation lens and suction system (LenSx, Alcon and Victus, Bausch and Lomb) or a fluid filled suction ring (Catalys, Johnson and Johnson and LensAR) [70]. The systems mentioned have all been CE marked and approved by the USA Food and Drug Administration for cataract surgery (Table 1).

The main advantages of fs-assisted cataract surgeries are the precision and repeatability of laser incisions to the tissue; reduction in ultrasound energy used for emulsification (liquification) of the lens nucleus by precutting it into small pieces; perfect sizing of corneal incision with regards to position, length and depth; and predictability in capsulotomy size and position. Despite the aforementioned obvious advantages and numerous studies showing superiority in performing the single surgical steps over the ones manually performed by surgeons, meta-analysis studies could not prove overall outcome advantages of fs-laser-assisted surgery versus the conventional phacoemulsification manual operation [71,72]. A randomized multicenter clinical trial including 1476 eyes of 907 patients could not prove superiority of fs-assisted cataract surgery over the traditional phacoemulsification method [73]. Nevertheless, the authors report no severe adverse events during the fs-laser procedure. Similarly, review articles emphasize usefulness of fs-assisted cataract surgery in some patient groups, i.e., those with low corneal endothelial cell counts, but a clear advantage of the fs method over manual phacoemulsification is not reported in routine cases [74,75]. Furthermore, the authors question the cost-effectiveness of fs-assisted surgery.

The question remains: Why should a high-precision system not be superior to the manual? Two answers merit consideration. (1) Several studies were done using the first-generation femtosecond laser systems comparing those to the conventional phacoemulsification surgery that has undergone evolution and perfection for several decades. Results of comparative studies using the newest laser devices could give a better comparison. (2) In most comparative studies, the conventional surgery has been performed by outstanding, high-volume, and exceptionally talented cataract surgeons. Comparing the visual outcomes of such surgeons to the machine results is like assessing the abilities and outcomes of first-generation autonomous driving systems to those of Formula-1 drivers. In the following, we describe each step of the cataract surgery taken over by the fs machine.

#### 3.3.1. Capsulotomy

Traditionally, cataract surgeons access the cataractous lens by manually opening the anterior lens capsule by pulling in a continuous curvilinear manner. This maneuver is performed using a needle or forceps and is called capsulorhexis. While the hard inner part of the lens is removed, the outer capsular bag is maintained. This bag is used as a mounting plate to fixate an implanted acrylic intraocular lens at the same position from where the cataractous lens was removed. Size, position, and shape of the capsulorhexis are related to the effective lens position, a determinant of the intraocular lens (IOL) power. The IOL power determines the postoperative refractive error of the eye. Inappropriate sizing of the capsular opening may result in IOL tilt, decentration, and increased posterior lens capsule opacification [76,77,78]. Perfect lens position is of particular importance to the IOLs with complex optical properties, e.g., multifocal, toric (for astigmatism correction), or those with extended depth of focus [79,80].

While a perfect capsular opening is of the highest significance for surgical success, it is one of the most challenging maneuvers in cataract surgery. The learning curve of surgeons in training can be quite flat. At the same time, even in the hands of the most experienced surgeons, the capsulorhexis can be unpredictable and perfect sizing at the submillimeter precision appears only possible with machines performing the step. Furthermore, the manual outcome is dependent on the axial length of the eye, pupil size, image enlargement by the cornea [81], and inherent features of the individual eye, for example, true exfoliation of the lens capsule (an eye disease) [82]. Femtosecond lasers overcome all these challenges by creating precise, predictable, repeatable, well-centered capsular opening, called laser capsulotomy, even in challenging cases with loose zonules (tissue fibers holding the lens in place), pediatric or mature cataracts, shallow anterior chamber, etc. Machine superiority has been demonstrated in several studies [78,83]. Another potential advantage of laser capsulotomy is centration of the opening on the eye’s true optical axis or the lens apex instead of centration on the pupil, as is usually done in manual capsulorhexis. Furthermore, innovative IOLs are available that are dependent on perfect capsulotomy sizing at a submillimeter level. Those designs allow IOL centration based on the capsulotomy rather than on the capsular bag [84].

#### 3.3.2. Nucleus Fragmentation

The human lens loses transparency and flexibility throughout life. A cataractous lens cannot be removed through a small incision by suction alone; rather, emulsification or fragmentation of the hard lens nucleus is necessary. Conventional cataract surgery uses ultrasound energy within the eye to liquefy the lens nucleus. Femtosecond laser technology allows precutting the nucleus in almost any imaginable shape and reduces the ultrasound energy needed for emulsification. This is an advantage as the ultrasound energy is a cause of oxidative stress, heat, and inflammation, and damage to the tissue [85]. The most susceptible tissue is the one-layer cell sheet of the corneal endothelium, which is of utmost importance to corneal transparency. Studies have shown less endothelial cell loss and decreased corneal swelling when using fs technology [86,87,88]. The importance of protection to the corneal endothelium becomes clear when we consider that corneal endothelial cells do not multiply after injury. The only way to repair the damage once clinically relevant is lamellar corneal grafting (transplantation). Several lens fragmentation patterns have been introduced, and currently, it is not yet clear which design to prefer in a particular clinical setting. Future studies should focus on the optimal fs lens nucleus fragmentation pattern to reduce effective ultrasound energy used intraoperatively.

#### 3.3.3. Corneal Incisions 

Full-thickness incisions

Full-thickness incisions through the cornea are necessary for the introduction of instruments into the eye. Traditionally, a metal scalpel or diamonds are used for creating them in different sizes. Fs technology allows predictable sizing (width, length, and depth) of full-thickness corneal incisions. Since these incisions need to be self-sealing, a perfect wound architecture incision is mandatory to prevent wound leakage postoperatively. Incorrect positioning of the wound induces astigmatism and can provoke prolapse of the iris during the surgery. Studies have shown increased repeatability and safety of wound construction using fs technology, resulting in higher stability and water tightness [89,90,91].

Partial-thickness incisions

Partial-thickness incisions into the cornea help to reduce preoperatively existing corneal astigmatism. Several nomograms have been developed, addressing length, position, and depth of the incisions for different amounts of astigmatism. Fs technology allows higher predictability and repeatability of partial thickness incisions, or even completely intrastromal corneal incisions [50]. Since the incision depths are up to 90% of the corneal thickness, laser precision helps prevent inadvertent penetration, as reported in manually performed antiastigmatic keratotomies. Fs-laser-assisted corneal incision could be as safe and effective as toric IOLs to reduce astigmatism [92]. Future studies will show whether predictability, safety, and efficacy of fs-laser-assisted keratotomies can be further improved by implementing nonperpendicular incision directions.

#### 3.3.4. Future Applications

Probably the most important evolutionary step in ophthalmic fs-laser devices will be miniaturization, mobility, and versatility. Tools available soon are fs laser-assisted primary posterior capsulotomy and lens capsule marking for positioning of toric IOL. On the horizon, another technology involves changing the IOL power postoperatively through fs-laser energy to achieve emmetropia in all eyes [93].

## 4. Summary

In summary, fs-laser technology has evolved over the past decades into a precise and reliable tool in ophthalmic surgery. While some of the applications have not finished evolving and require further research and development, fs-laser-assisted cataract and corneal surgery have reached highly standardized levels worldwide. For these surgeries, fs-laser technology has improved patient safety and clinical outcomes and opened gateways to new surgical approaches.

## Figures and Tables

**Figure 1 micromachines-12-00122-f001:**
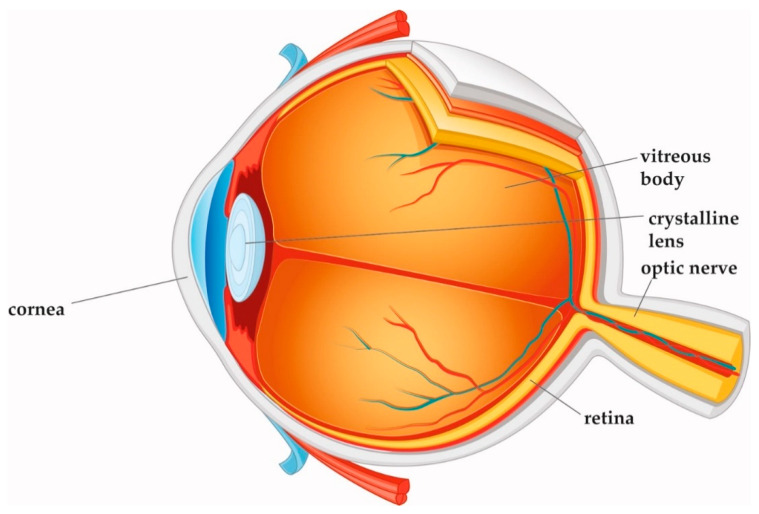
Cross-section of the eye. Cornea, crystalline lens, and vitreous body are transparent in the healthy eye.

**Figure 2 micromachines-12-00122-f002:**
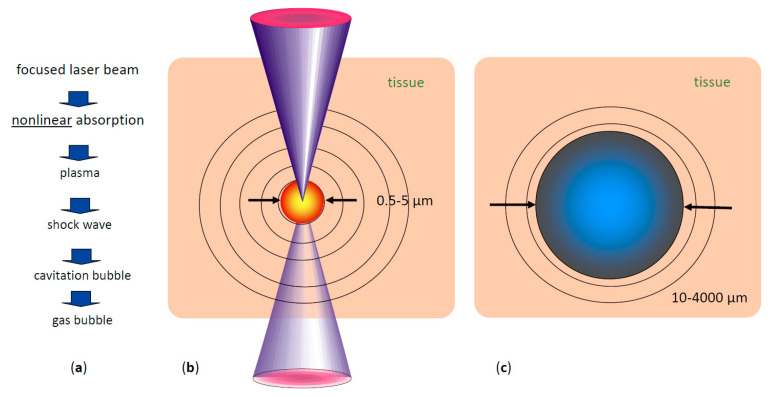
Short pulse laser effects in tissue: (**a**) sequence of effects and induced events, (**b**) plasma size range and pressure wave pattern, (**c**) range of possible cavitation bubble dimensions (pulse energy-dependent) [9].

**Figure 3 micromachines-12-00122-f003:**
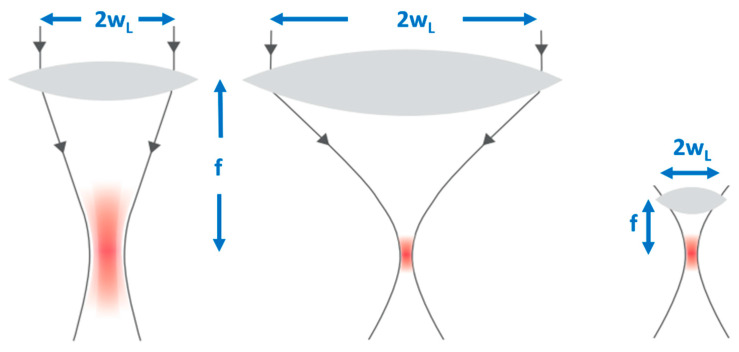
The focal volume of a Gaussian laser beam scales with the numerical aperture NA = w/f of the focusing lens. The larger the NA, the smaller the focal spot volume.

**Figure 4 micromachines-12-00122-f004:**
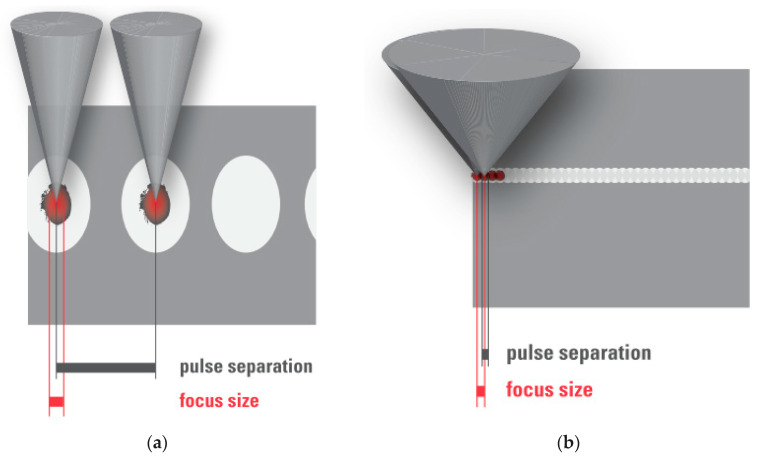
(**a**) High pulse energy, low repetition rate (large spot separation); (**b**) low pulse energy, high repetition rate (small spot separation, overlapping plasma effects of spots).

**Figure 5 micromachines-12-00122-f005:**
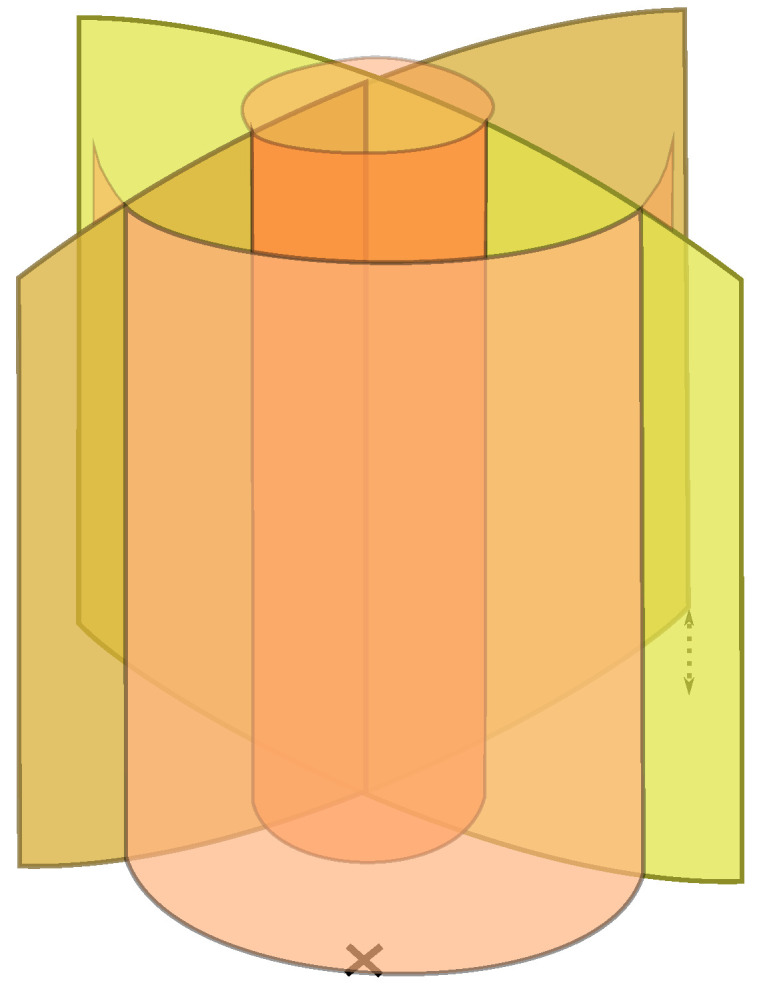
Three-dimensional laser focus scan pattern used for the fragmentation of cataractous lenses.

**Figure 6 micromachines-12-00122-f006:**
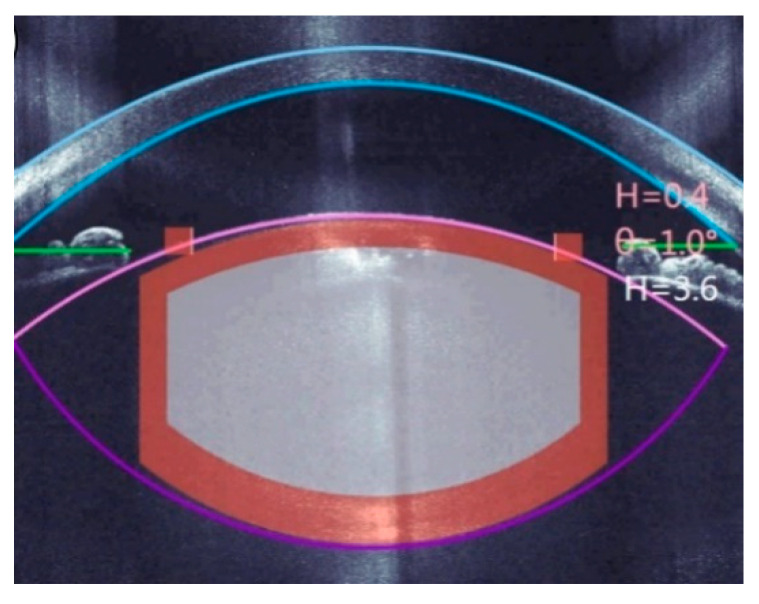
Example of the optical coherence tomography (OCT)-guided placement of an fs-laser cut pattern (blue: corneal anterior and posterior surface; pink and purple: lens anterior and posterior surface; green: iris plane).

**Figure 7 micromachines-12-00122-f007:**
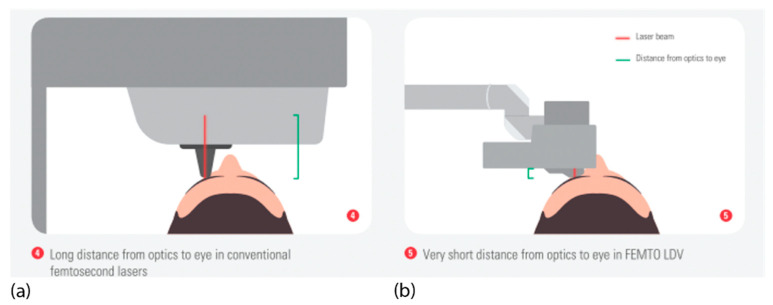
Typical eye docking methods of fs lasers: (**a**) head under fixed laser housing, (**b**) articulated arm with handpiece placed onto the eye; green: distance of eye surface to laser optics.

**Figure 8 micromachines-12-00122-f008:**
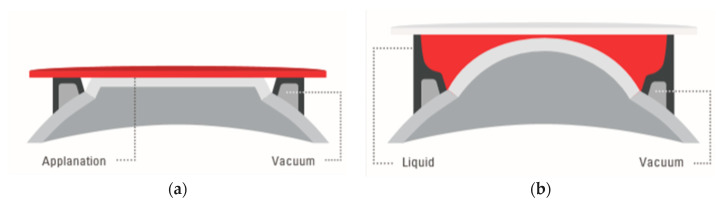
Typical patient interface designs: (**a**) contact interface in direct touch with the cornea (flat or curved), and (**b**) liquid optics interface, no direct touch on the cornea, no deformation.

**Figure 9 micromachines-12-00122-f009:**
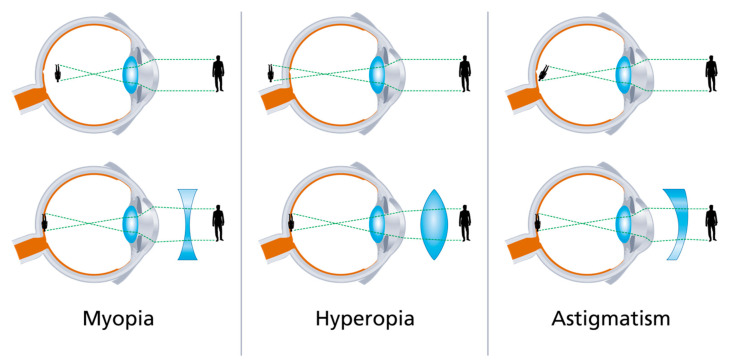
Illustration of different types of refractive error and their correction with lenses. Corneal refractive surgery changes the shape of the cornea according to the corrective lenses.

**Figure 10 micromachines-12-00122-f010:**
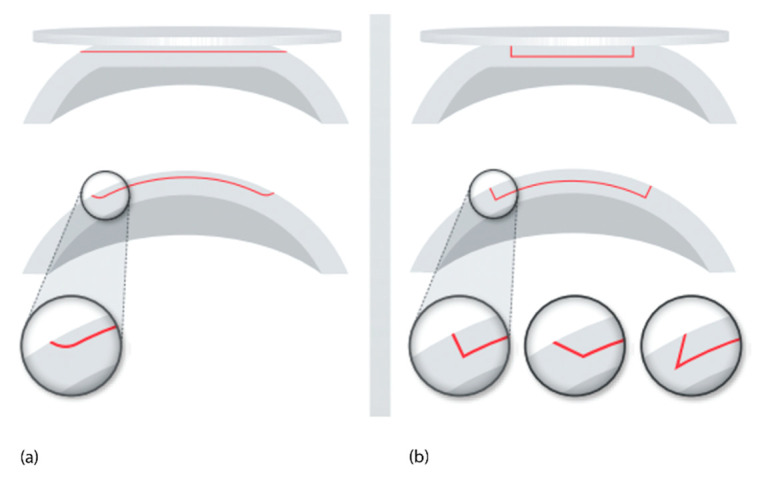
Corneal flaps cut by fs laser: (**a**) straight plane (red) with continuously curved sides cut during vacuum docking to a flat interface, (**b**) angulated side cut options (3D cutting geometry).

**Figure 11 micromachines-12-00122-f011:**
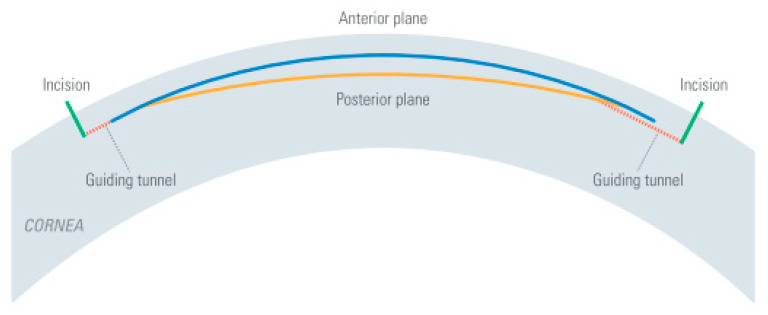
Schematic view of intrastromal lenticule cuts performed by an fs laser. The lenticule created between the anterior (blue) and posterior (yellow line) cut planes is extracted by the surgeon via an incision (green line). Optionally there is a second incision created to help mobilize the lenticule.

**Figure 12 micromachines-12-00122-f012:**
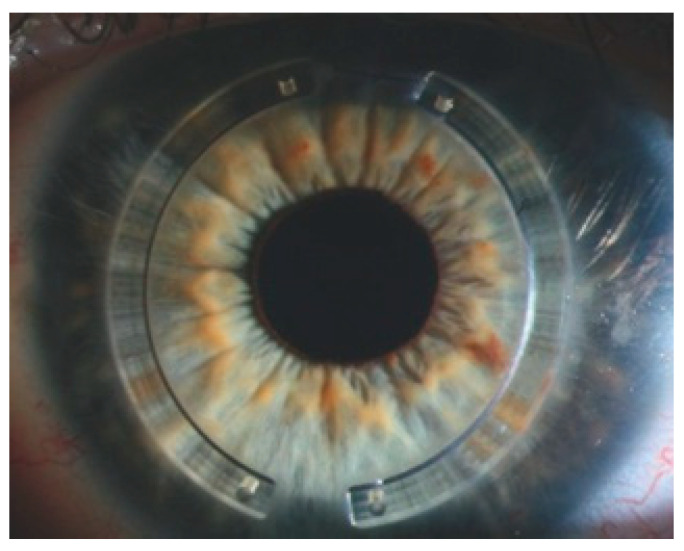
Intrastromal corneal ring segments implanted into fs-laser cut pockets.

**Figure 13 micromachines-12-00122-f013:**
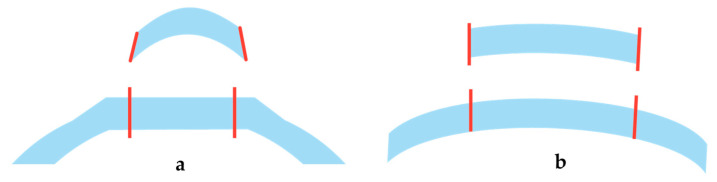
Comparison of donor and recipient trephination profiles: (**a**) applanation, (**b**) liquid optic interface.

**Figure 14 micromachines-12-00122-f014:**
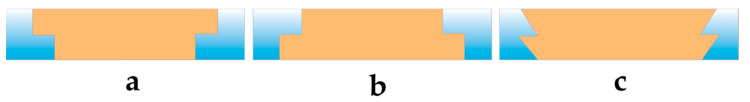
Sidecut profiles: (**a**) mushroom, (**b**) top hat, (**c**) zig-zag.

**Figure 15 micromachines-12-00122-f015:**
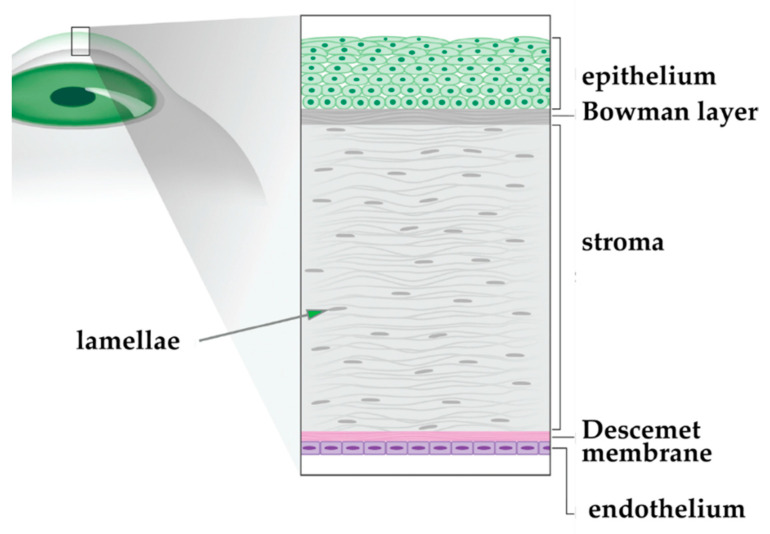
Illustration of corneal layers.

**Figure 16 micromachines-12-00122-f016:**
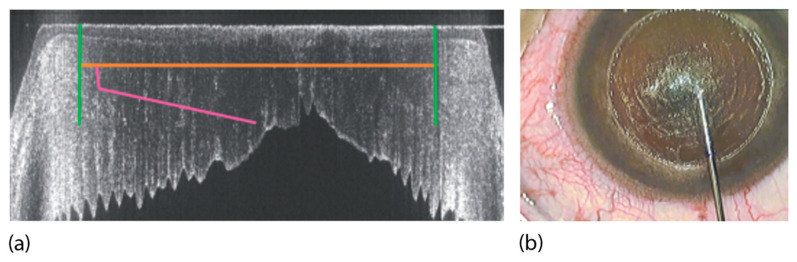
Deep anterior lamellar keratoplasty (DALK) procedure: (**a**) OCT-guided placement of the cuts: side cut in green, pre-Descemet’s stroma cut in orange, guiding channel in pink. The posterior cornea is folded due to the applanating docking. (**b**) Insertion of the cannula for air injection through the precut guiding channel, just above the Descemet’s membrane.

**Table 1 micromachines-12-00122-t001:** Overview of five commercially available femtosecond lasers for eye surgery.

	IntraLase (AMO, USA)	Wavelight FS200 (Alcon, USA)	LenSx (Alcon, USA)	LensAR (LensAR, Topcon, USA)	Catalys (Johnson and Johnson, USA)	Victus (Bausch and Lomb, Germany)	VisuMax (Zeiss Meditec, Germany)	LDV Z8 (Ziemer, Switzer-land)	Atos * (Schwind, Germany)
Pulse repetition rate (kHz)	30–150	200	60	80	120	80/160	500	10,000	<to 4000
Pulse duration (fs)	>500	350	600–800	500	<600	290–550	220–580	250	<295
Pulse energy (µJ)	Ca. 1	<1.5	>15	7–15	3–10	6–10	<1	<1	<1
Applications:									
LASIK flaps	x	x		x		x	x	x	x
Refractive Lenticules							x	x	x
Cornea Surgery	x	x				x		x	
Cataract Surgery			x	x	x	x		x	
Patient interface	Flat applan. interface	Flat applan. interface	Curved softfit interface	Fluid filled interface	Liquid interface	Semiliquid curved interface	Curved interface	Liquid and flat interfaces	Curved interface

*: according to manufacturer.

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
