# Peer review of "Femtosecond-Laser Assisted Surgery of the Eye: Overview and Impact of the Low-Energy Concept"

_micromachines, 2021, doi:10.3390/mi12020122_

Round 1

Reviewer 1 Report

General comments:

It was interesting reading the reported study. The study gives a very good overview of the application of the femtosecond laser, but more technical details would certainly be interesting. Furthermore, it is noticeable that one laser system is slightly preferred in the article.

Specific comments

Line 67: There is talk of a cavitation bubble here. Can the authors write something more specific about how the cavitation bubble comes about, what its contents are and with what the size of the cavitation bubble correlates?

Line 97: The initial version of the Intralase had a repetition rate of 15 kHz and not 30 kHz. This version came later as an upgrade.

Line 130: It is written that with the high-energy laser systems, the cutting process is achieved mechanically through bubbles. In the case of the low-energy laser, no explicit reference is made to how this occurs, but only to the overlapping laser spots. It would be good if an explicit description were given for the low-energy laser of how the cutting process occurs.

Line 174: In OCT, we speak of the Fourrier domain. For the sake of completeness, the initial OCT technology TD (time domain) should be mentioned. Furthermore, the technical advantage of FD should be discussed. As a further point, the FD is divided into 2 further technologies, a spectral-domain and swept-domain. Here, the advantages and disadvantages of the OCTs should be briefly discussed and in which device they were installed.

Line 188: With the Lensar laser, the imaging is solved by 3-D confocal structured illumination with automatic biometry, Scheimpflug-based. In a review article, its description is also included to get a comprehensive overview of the applied technologies.

Line 192: The stability of the docking system is very important for its application. The most unfavourable thing during an application is a vacuum loss. Here you should briefly point out how this problem is counteracted by the manufacturers.

Line 261-264: Can the authors provide a literature reference for each of these postulated statements, as individual points are controversially discussed in the literature.

Reviewer 2 Report

I have read the article by Latz et al. on the use the femtosecond lasers in eye surgery. This is a well written and interesting review.

Comments:
- as the authors analyze the use of femtosecond laser, it would be beneficial to shortly present laser-tissue interaction (but not only for fs-lasers, as is done in 2.2)
- 2.1.1. Nd:YAG lasers are presented, however clinnical applications are not stated
- Figure 8 presents patient interface designs. It would be beneficial to present at this point (or in a separate Table or in modified Table 1) which laser models use which interface.
- what are the benefits and drawback of the discussed patient interfaces? (e.g., this was partially presented here: 10.3928/1081597X-20190710-02)
- the table 1 should be significantly extended. It should present fs-lasers not only used in cataract surgery, but also for refractive surgery. Some lasers can be used for both types of surgery - the applications for every device should be presented. Moreover, all currently available lasers should be stated. How about the brand new Schwind Atos?
- using fs-lasers for keratoplasty should be shortly compared with the literature about excimer lasers...
- what is the main limitation of partial thickness incisions? Does it provide a long-term effect?
- how about problems with using smile for other refractive errors then myopia? This could be the main limitation of this tehchnique and should be discussed...
- OCT studies have presented that although fs-cuts are more precise, they do not provide long-term wound adherence (Rodrigues R, Santos MSD, Silver RE, et al. Corneal incision architecture: VICTUS femtosecond laser vs manual keratome. Clin Ophthalmol 2019; 13: 147–152. / Chaves de Medeiros AL, Vilar CMC, Magalhaes KRP, et al. Architecture evaluation of the main clear corneal incisions in femtosecond laser-assisted cataract surgery by optical coherence tomography imaging. Clin Ophthalmol 2019; 13: 365–372.)
- Currently, there are several articles on FLACS analyzing results of large studies (e.g., a review 10.22608/APO.2017159, a more one review 10.1177/1120672120922448 or finally the results of the FEMCAT study recently published in the Lancet). These studies should be cited and discussed

Round 2

Reviewer 1 Report

I would like to congratulate the authors for a very nice review paper.